# Determining Ancestry between Rodent- and Human-Derived Virus Sequences in Endemic Foci: Towards a More Integral Molecular Epidemiology of Lassa Fever within West Africa

**DOI:** 10.3390/biology9020026

**Published:** 2020-02-07

**Authors:** Ayodeji Olayemi, Adetunji Samuel Adesina, Thomas Strecker, N’Faly Magassouba, Elisabeth Fichet-Calvet

**Affiliations:** 1Natural History Museum, Obafemi Awolowo University, Ile Ife, HO220005, Nigeria; aolayemi@oauife.edu.ng; 2Department of Biochemistry and Molecular Biology, Obafemi Awolowo University, Ile Ife HO220005, Nigeria; asadesina@yahoo.com; 3Institute of Virology, Philipps University, 35043 Marburg, Germany; strecker@staff.uni-marburg.de; 4Laboratoire des Fièvres Hémorragiques Virales, Conakry, Guinea; cmagassouba01@gmail.com; 5Department of Virology, Bernhard-Nocht Institute of Tropical Medicine, 20359 Hamburg, Germany

**Keywords:** Lassa virus, zoonosis, *Mastomys*, transmission, humans, West Africa, phylogeny, emergence

## Abstract

Lassa fever is a viral hemorrhagic illness responsible for thousands of human deaths in West Africa yearly. Rodents are known as natural reservoirs of the causative Lassa mammarenavirus (LASV) while humans are regarded as incidental, spill-over hosts. Analysis of genetic sequences continues to add to our understanding of the evolutionary history, emergence patterns, and the epidemiology of LASV. Hitherto, the source of data in such investigations has mainly comprised human clinical samples. Presently, a rise in the quantity of virus strains accessed through ecological studies over the last 15 years now allows us to explore how LASV sequences obtained from rodents might affect phylogenetic patterns. In this study, we phylogenetically compared LASV sequences obtained from both rodents and humans across West Africa, including those from two localities highly endemic for the disease: Ekpoma in Nigeria and Kenema in Sierra Leone. We performed a time-calibrated phylogeny, using a Bayesian analysis on 198 taxa, including 102 sequences from rodents and 96 from humans. Contrary to expectation, our results show that LASV strains detected in humans within these localities, even those sampled recently, are consistently ancient to those circulating in rodents in the same area. We discuss the possibilities connected to this preliminary outcome. We also propose modalities to guide more comprehensive comparisons of human and rodent data in LASV molecular epidemiological studies.

## 1. Introduction

Lassa fever causes a deadly zoonotic disease that kills 5000 or more people every year in West Africa [1]. The Lassa mammarenavirus (LASV), which causes this illness, is borne by rodents. The Natal multimammate mouse *Mastomys natalensis* is recognized as the main LASV reservoir while the Guinea multimammate mouse *M. erythroleucus* and the African wood mouse *Hylomyscus pamfi* were recently identified as additional hosts [2,3]. Primary (rodent-to-human) transmission of the virus occurs when humans come in contact with excreta or body fluids of infected rodents. Human-to-human spread is regarded as secondary. Further knowledge on the epidemiology of Lassa fever, particularly at the rodent–human interface, is needed to control this disease.

The recent surge in genetic sequencing has helped to advance our understanding concerning the evolutionary history of LASV. Geographic distribution and estimated emergence dates have been described for four LASV lineages: I–III in Nigeria and IV in Guinea, Sierra Leone, and Liberia [4]. Additional putative lineages were recently detected in southern Mali, northern Ivory Coast, and Togo [5,6,7].

In addition, a generation of LASV sequences at the local scale during outbreaks within specific countries such as Nigeria and Sierra Leone contributes to the molecular epidemiology of the virus [8,9,10]. Phylogenetic analyses of these sequences, dominated by clinical samples from humans, commonly depict independent chains of transmission in humans originating from rodents. However, an increasing number of rodent-derived sequences from ecological studies targeted at endemic localities where humans have been regularly sequenced now provide the opportunity to further examine primary LASV transmission [3,11,12,13,14].

During outbreaks of rodent-borne viral hemorrhagic fevers, phylogenetic comparison has helped to link infection in human patients to rodent populations for hantaviruses in the USA [15], western Bolivia and Paraguay [16], Finland [17], and Germany [18]. This is also similar for arenaviruses, such as the Lymphocytic choriomeningitis mammarenavirus in France [19] and LASV in southern Mali [6]. These studies typically involved few virus sequences obtained both from the infected patients and rodents in their domestic surroundings, and do not allow a statistical comparison between rodent versus human hosts.

Here, we seek to phylogenetically infer ancestry and descent between LASV sequences detected in rodents and humans in selected localities within West Africa in order to provide increased insight into virus transmission at the rodent–human boundary. Our aim was to compile a dataset including LASV sequences collected from both humans and *Mastomys* rodents within distinctive hotspots of Lassa fever. Two localities in West Africa currently fit this description: Kenema (represented by Kenema and other closely surrounding localities within a 35 km radius, Sierra Leone) and Ekpoma (comprising Ekpoma and Eguare Egoro, only 6 km from each other within Nigeria).

## 2. Materials and Methods

### 2.1. LASV Sequences in Mastomys

We used complete LASV sequences from *Mastomys* spp. trapped in Guinea (*n* = 5), Mali (*n* = 4), Nigeria (*n* = 2), and Sierra Leone (*n* = 10) [3,6,8,20,21]. Since these full genomes are few in number, we supplemented them with partial sequences from other *Mastomys* trapped in Guinea and Nigeria. In Guinea, we sequenced the GP 1kb portion to complement NP sequences published in Lecompte et al. [12] and Fichet-Calvet et al. [11]. A fragment of 950 nucleotides was amplified using a RT-PCR protocol with the following primers: LVS 36-fwd (ACCGGGGATCCTAGGCATTT) and OWS 1000-rev (AGCATGTCACAGAAYTCYTCATCATG) for the strains from Faranah and Denguedou. In Nigeria, we sequenced the NP to complement LASV GP sequences published in Olayemi et al. [14] from *Mastomys natalensis* that were sampled during 2011–2012. We designed primers to amplify the NP at the same position of those published in Bowen et al. [4]. A fragment of 800 nucleotides was amplified using a RT-PCR protocol (Appendix A) with the following primers: LVSnig 1669-fwd (TATATTGAGTCCTCCTGACACAG) and LVSnig 2511-rev (TGTTGGAGACCATCAAGGTT) (Appendix A). Furthermore, to expand the timeframe of our collection in Ekpoma, we added LASV GP 1kb and partial NP sequences from 13 *M. natalensis* trapped during 2014–2016. All PCR fragments were sequenced on both strands. The sequences were assembled and aligned using the Mac Vector software (Mac Vector 16.0.8, Inc, Apex North Carolina 27502, USA). In total, we generated 46 partial GP sequences of LASV from *M. natalensis* in Faranah and in Denguedou; 13 partial NP from *M. natalensis* in Ekpoma collected in 2011–2012; and GP 1kb plus partial NP from 13 *M. natalensis* also trapped in Ekpoma during 2014–2016 (Appendix A). Sequences were submitted to GenBank under accession numbers MN123651-MN123709 and MT001745-MT001770. The sequences from *Mastomys* used in our study are listed in Appendix A.

### 2.2. LASV Sequences in Humans

The sequences of LASV isolated in humans are readily available in Genbank, thanks in particular to the research of Andersen et al. [8] and Siddle et al. [10]. They mainly concern lineages II and IV circulating around Ekpoma in Nigeria and Kenema in Sierra Leone, respectively. In this article, we use the new Nigeria sequences published in 2018 because they are localized by state. For Sierra Leone, however, we use those of 2015 [8] differentiating sequences from Kenema, Bo, and Liberia. The sequences from humans used in our study are listed in Appendix A.

### 2.3. Phylogenetic Analysis

The phylogenies were inferred by the Bayesian Markov Chain Monte Carlo (MCMC) method implemented in BEAST software (https://beast.community) [22]. 

Model 1: To evaluate the sequences from humans living near Kenema only, we did a preliminary analysis gathering all the known LASV strains, including those from known areas. In BEAUTI, the parameters are:One partition, NP full sequence for 226 taxa;Substitution model as GTR + gamma without codon partition;Lognormal relax clock;Coalescent tree with a constant size population;MCMC = 10 M, echo states, and log parameters every 10,000.

It was therefore possible to remove the samples coming from Bo (G1442, G1529, G2141, G2230, G2259, G2363, G2431, G2557, G2615, G2723, G2789, G3034, G3278) and those from Liberia from the Kenema dataset. These samples correspond to patients tested in Kenema but coming from neighboring areas.

Model 2: To get an estimation of the time of divergence, with both humans and *Mastomys*, we merged the GP and NP in a combined phylogenetic analysis. As we were searching for LASV diversity, all similar sequences were removed.

In BEAUTI, the parameters are:Two partitions, GP 900 nt and NP 712 nt for 198 sequences. The substitution models, clock, and trees are linked;Eight taxa were defined: *Homo* Ekpoma, *Homo* Kenema, *Mastomys* Denguedou, *Mastomys* Ekpoma, *Mastomys* Faranah, *Mastomys* Kenema, *Mastomys* Madina, and *Mastomys* Mali;Tip dates at the nearest day;Substitution model as GTR + gamma and codon partition with positions 1,2,3;Strict (model 2a) or uncorrelated relaxed (model 2b) or random local (model 2c) or fixed local (model 2d) clock;Coalescent tree with a constant size population;MCMC = 50 M, echo states, and log parameters every 50,000.

The xml files issued from BEAUTI were run in BEAST, the log files checked in TRACER, and consensus trees were visualized through FigTree (BEAST packages, https://beast.community/programs).

## 3. Results

### 3.1. LASV in Mastomys

The six viral populations isolated from *M. natalensis* and *M. erythroleucus* show very different emergence periods according to the rodent populations from which they originated in West Africa (Figure 1a, Table 1). Indeed, the LASV clade obtained from rodents in Mali is the oldest and dates back to 96 years, whereas that obtained in Denguedou near the Sierra Leonean border is very recent and seems to have emerged 20 years ago. Between these two emerged the LASV population of Faranah in Upper Guinea; those from Ekpoma and Kenema synchronously; and then that of Madina Oula in coastal Guinea (Figure 1b). The pattern is similar regardless of clock model; strict or uncorrelated lognormal (Table 1). We did not present the results of the random and fixed local clock because the models never converged well, i.e., the ESS parameters did not reach the value of 100 after a MCMC chain of 50 M. Figure 1c shows the phylogenetic tree gathering the rodent- and human-derived LASV sequences in West Africa. 

### 3.2. LASV in Homo

To compare the sequences obtained from humans to those from rodents, we have only two sites with sufficient numbers of individual human-derived sequences for analysis: Ekpoma in Nigeria and Kenema in Sierra Leone. This analysis is summarized in Figure 2 and Table 1 where we note the same pattern: the sequences observed in *Homo* are on average earlier than those detected in *Mastomys*. A superimposition exists between the two viral populations, which indicates a close contact and the passage of the virus from one host to another. On the Kenema graph (Figure 2b), the two peaks are more disjointed than in Ekpoma (Figure 2a), probably because of a greater distance between the sites sampled within this locality. Additional analyses that do not take into account cases from Bo or Liberia show even more disjointed peaks (data not shown).

## 4. Discussion

### 4.1. LASV Emergence Per Locality

In the absence of the complete S segment, a combination of partial GP and NP sequences to increase genome information offers a good compromise (1612 nt together in one analysis instead of two separate analyses with GP, 900 nt, and NP, 712 nt, respectively). This combination with two partial genes in the open reading frame has never been done before. However, our analysis showed more recent dates of divergence compared to studies done on the complete S segment. For instance, the date of divergence in Sierra Leone is 107 (91–124) years in our analysis versus 118 (102–137) in the Andersen study [8]. Divergence out of Nigeria is 249 (212–284) years in our study versus 283 (241–331) described by Andersen et al. We therefore suggest that the dates presented in this analysis are slightly underestimated because we used a shorter nucleotide sequence than the complete S segment (1612 nt instead of ±3400 nt). As we consider that the bias is similar for each strain, the comparison between the different viral populations remains valuable.

The viral population from Malian rodents is the oldest (96 years), and we can suggest that the virus was then exported to Faranah, South Upper Guinea, 20 years later, and lately to Sierra Leone with another jump of 20 years. The viral population found in Denguedou, a locality between the Sierra Leonean/Guinean border and Guekedou, is probably a signature of human displacements during the civil war in Sierra Leone (1994–2004) as it emerged only 20 years ago. During that time, 700,000 Sierra Leonean refugees arrived in Forest Guinea and were dispatched all along the border. The times of the most recent common ancestor (tmrca) estimated in our study is higher than the one estimated in Lalis et al. [23] because we used the GP in addition to a short NP fragment. The recent viral population discovered in Madina Oula in *M. erythroleucus* [3] emerged between those observed in Denguedou and those observed in Kenema. These sequences are phylogenetically very different as they cluster to Liberian strains. The village of Madina Oula regularly receives some loggers from Forest Guinea who probably imported this strain. Finally, the strains found in the Nigerian *Mastomys* from Ekpoma are quite recent, ±55 years, similarly to those found in the Kenema area.

### 4.2. LASV Emergence Per Host

Phylogenetic clustering in the clades representing Kenema (within lineage IV) and Ekpoma (in lineage II) show that human-derived LASV sequences are significantly ancestral to those obtained from the *M. natalensis* rodent, regardless of time of sampling. Normally, one would have expected rodent-to-human transmission (where long-term evolution of the virus takes place in rodents and spills over intermittently to humans) to be consistent with the LASV strains from *M. natalensis* being generally ancestral or at least randomly interspersed on the phylogenetic tree compared to those obtained from humans [13]. Therefore, our results may suggest the occurrence of human-to-rodent transmission (reverse zoonosis).

During epidemics of certain diseases, patterns of ancestry and descent have helped determine that virus routes of transmission were from humans to other mammals. For instance, investigations in Sri Lanka and Mexico concerning Influenza A (H1N1 and H3N2) emergence in a cluster of farms showed that virus transmission between humans preceded infection in swine [24,25]. Other diseases such as rabies, hepatitis E, and respiratory syndromes were recently proved to be transmitted by humans to dogs, raw pork, and chimpanzees [26,27,28,29]. The idea that humans may function in some circumstances as “reservoir hosts” of LASV is counter-intuitive, as infection is usually associated with acuteness, severe illness and mortality. However, there are also data demonstrating that up to 70% of LASV seropositive humans show benign or no symptoms [1]. Moreover, those recovered from the disease may continue to shed the virus in urine or saliva for up to 60 days [1,30], contaminating domestic surfaces, waste, and sewage systems accessible to rodents.

We are aware that the notion of reverse zoonosis in Lassa fever would require a large body of additional evidence to become established, including experimental approaches. Furthermore, as we envisage the impending availability of a much expanded pool of rodent-sourced LASV sequences against which our findings can be tested, we regard our research as a precursory, yet important foundation that will inform subsequent data collection and analyses.

### 4.3. Future Sampling

Appropriateness and robustness in data acquisition lend to interpretability. In Ekpoma, for instance, which is known as one of the most endemic localities for Lassa fever in Nigeria [31], we conducted rodent trapping in 24 GPS-tracked sites spanning the city, including precise addresses where Lassa fever cases were previously recorded. We sampled these sites twice each year during 2011–2012 (Olayemi et al. [14]) and in eight sessions through 2014–2016. Therefore, we find it surprising that, considering the geographic and temporal scope of our compilation, almost all the LASV strains we sequenced from *M. natalensis* were phylogenetically younger in comparison to those obtained from humans.

In addition to the structured sampling in the ecological projects that generated most of the rodent-derived LASV sequences, we suggest conducting opportunistic collection approaches in future studies. In such a case, *Mastomys* would be captured in the homes of Lassa fever patients as soon as they are diagnosed. This would result in a greater probability of detecting the older strains that have already been documented in humans; if they are indeed circulating in rodents. Studies in molecular epidemiology such as ours can be notoriously confounded by misleading human geographic data (due to movement after infection or poor hospital records [13]), causing virus strains from out-of-town to be labeled for a certain locality; and, in turn, influencing the interpretation of ancestry and descent for strains in that locality.

Thus, future studies should aim to link human and rodent data more directly at a finer scale (i.e., at the level of the home address), producing multiple, comparable phylogenetic reconstructions of who-infected-who within a locality. This will help to further support our findings and will provide much needed insight underlying LASV transmission mechanisms at the rodent–human interface.

## 5. Conclusions

Our findings do not necessarily contradict the rodent-to-human route of LASV transmission, which is supported by recent LASV phylogenies based on sequences mostly sampled from humans [8,9,10]. Instead, we call for a broader paradigm including infection in the reverse direction. In such a scenario, few infectious humans could initiate a LASV focus area, which is then perpetuated by commensal rodents acting as transmission enhancers.

## Figures and Tables

**Figure 1 biology-09-00026-f001:**
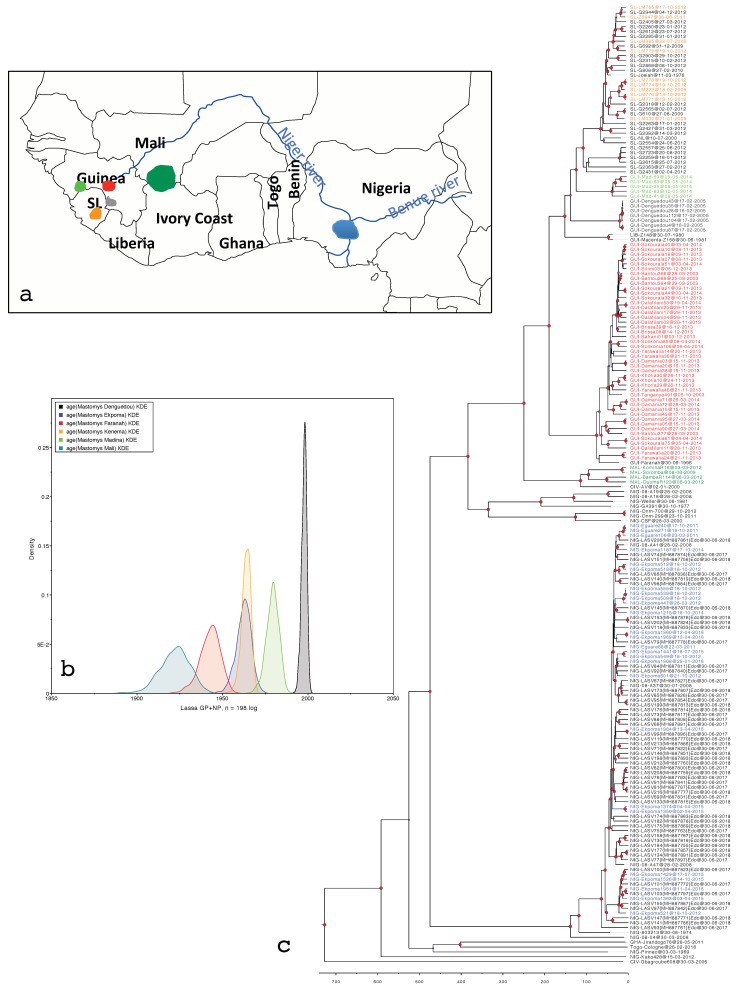
(**a**) Map of West Africa showing the geographic distribution of Lassa mammarenavirus (LASV) in *Mastomys* populations represented in the phylogenetic tree in the same colors. (**b**) LASV dates of emergence among these *Mastomys* populations (X axis). Density on the Y axis represents the probability of clock rate validity. (**c**) Phylogenetic tree featuring LASV sequences derived from humans (in black); from *Mastomys natalensis* in Ekpoma, Nigeria (blue); southern Mali (dark green); Kenema, Sierra Leone (orange); Denguedou, Forest Guinea (gray); Faranah, Upper Guinea (red); and from *M. erythroleucus* in Madina Oula, Guinea (light green). The tree combines GP and NP sequences analyzed in model 2a (strict clock, see Table 1). Red dots at the nodes indicate a posterior value of >0.9.

**Figure 2 biology-09-00026-f002:**
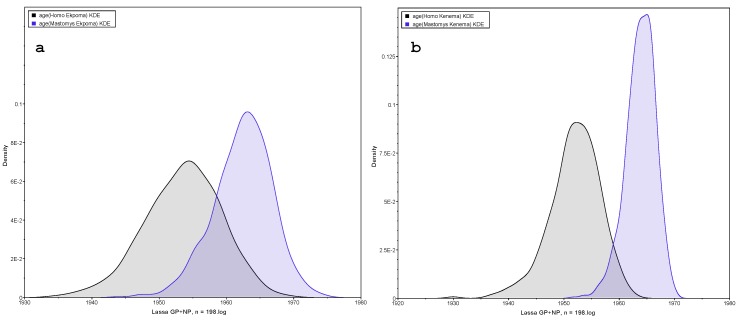
Dates of LASV emergence in *Mastomys natalensis* versus *Homo sapiens* living in (**a**) Ekpoma (Nigeria) and (**b**) Kenema (Sierra Leone). In both locations, the peak in humans shows the emergence of LASV earlier than the one in rodents. Peaks in humans include 58 sequences in Nigeria and 19 sequences in Sierra Leone. Peaks in rodents include 26 sequences in Nigeria and 10 in Sierra Leone (see Appendix A).

**Table 1 biology-09-00026-t001:** Times of the most recent common ancestor (tmrca) in years according to each model; 2a = strict clock, 2b = uncorrelated relaxed clock.

Taxon	Model 2a	Model 2b
*Homo* Ekpoma	65 (54–76)	63 (50–77)
*Homo* Kenema	66 (58–76)	66(55–77)
*Mastomys* Denguedou	20 (18–23)	20 (17–24)
*Mastomys* Ekpoma	56 (47–64)	54 (45–64)
*Mastomys* Faranah	75 (64–87)	71 (59–85)
*Mastomys* Kenema	54 (49–59)	53 (49–59)
*Mastomys* Madina	39 (32–46)	38 (29–47)
*Mastomys* Mali	96 (81–113)	89 (69–112)

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
