# Peer review of "Determining Ancestry between Rodent- and Human-Derived Virus Sequences in Endemic Foci: Towards a More Integral Molecular Epidemiology of Lassa Fever within West Africa"

_biology, 2020, doi:10.3390/biology9020026_

Round 1

Reviewer 1 Report

In this manuscritp, the authors compare LASV sequences obtained from both rodents and humans across West Africa. The results show that LASV strains detected in humans are consistently ancient to those circulating in rodents in the same area, indicating humans are likely more involved in maintaining and spreading LASV. However, the manuscript is not suitable for publication at present form.

Major comments:

The last two sentences are useless in the Abstract. Line 96, Table 1 should be Table 2. And therefore, Line 138, Table 2 should be Table 3. Move Table 1 and Table 2 into the supplemental files. Line 70, We downloaded complete LASV sequences from Mastomys spp., what do you mean? I think the word ‘downloaded’ here is not correct. The full genomes of LASV collected from these locations are few, which cannot represent the result. Extensive editing of English language and style required. Fig 2, In both locations, the peak in humans shows the emergence of 172 LASV earlier than the one in rodents, why? The conclusion that the virus was transmitted from human to animal only through sequence analysis is not rigorous enough, and experimental evidence is needed.

Reviewer 2 Report

The authors make use of additional partial sequencing of Lassa virus isolates from rodents and attempt to compare rodent and human sequences from similar regions and timepoints. Their data shows that the human isolates are consistently more ancient to those circulating in rodents. The authors come to the conclusion that humans are more involved in maintaining and spreading LASV than currently realized.

Major Comments:

-Unfortunately the study design is not capable of leading to the conclusion that there is more evolution occurring in humans than originally thought. While the data shows that the isolates are more ancient in humans there are many factors that were not controlled for in order to reach this conclusion. A large body of sequencing would need to be generated with isolates collected from humans and Mastomys in close proximity to one another and comparisons made in order to truly measure how ancient the isolates are with respect to one another. As presented by the authors the matching of samples from a large area during a large time period would not be sufficient given what we know about LASV diversity.

-The consistent low level of transmission person-person during outbreaks suggests that by and large humans are a dead end host, making it unlikely that a larger amount of evolution is occurring in humans suggested by the authors

-It is this authors opinion that the rodent sequencing data presented represents a strong contribution to the LASV field and should be published, however the current analysis and conclusions in there current format are unwarranted.

-I would recommend that the authors refrain from making any major conclusions the sequence comparisons and discuss the limitations of such comparisons more thoroughly in their discussion.

Minor Comments:

-the authors consistently refer to Mastomys natalensis as a "multimammate mouse". This is incorrect as Mastomys are a multimammate rat and should be fixed throughout the text.

-Both tables in the manuscript are labeled "Table 1" which should be fixed. I would also recommend placing those tables in the supplementary data.

-It wasn't clear if the new sequences were uploaded and provided to GenBank?

Round 2

Reviewer 1 Report

The comments were revised and answered in this revised manuscript properly. The revised manuscript is ready for publication now.

Reviewer 2 Report

The authors have done a good job revising and addressing my previous comments. I would recommend acceptance in its present form.